# Cost-effectiveness of livelihood interventions for families of children with cerebral palsy in rural Bangladesh

**Nuruzzaman Khan**[1,2‡], **Manik Chandra Das**[3,4,5‡], **Mahmudul Hassan Al Imam**[3,6,4,5], **Israt Jahan**[3,6,4,5], **Delwar Akbar**[7], **Mohammad Muhit**[4,5], **Nadia Badwai**[8,9] **Gulam Khandaker**[3,6,4,5,10,11] *

1 Department of Population Science, Jatiya Kabi Kazi Nazrul Islam University, Mymensingh, Bangladesh, 2 Nossal Institute for Global Health, Melbourne School of Population and Global Health, The University of Melbourne, Parkville, Victoria, Australia, 3 School of Health, Medical and Applied Sciences, Central Queensland University, Rockhampton, Queensland, Australia, 4 CSF Global, Dhaka, Bangladesh, 5 Asian Institute of Disability and Development (AIDD), University of South Asia, Dhaka, Bangladesh, 6 Central Queensland Public Health Unit, Central Queensland Hospital and Health Service, Rockhampton, Queensland, Australia, 7 School of Business and Law, Central Queensland University, Rockhampton, Queensland, Australia, 8 Cerebral Palsy Alliance, Sydney Medical School, The University of Sydney, Camperdown, Sydney, New South Wales, Australia, 9 Grace Centre for Newborn Intensive Care, Sydney Children's Hospital Network, Westmead, Sydney, New South Wales, Australia, 10 Discipline of Child and Adolescent Health, Sydney Medical School, The University of Sydney, Sydney, New South Wales, Australia, 11 Executive Director Medical Services, Central Queensland Hospital and Health Service, Rockhampton, Queensland, Australia

‡ These authors Joint first authors on this work.
* gulam.khandaker@health.qld.gov.au

## Abstract

### Background

Families of children with Cerebral Palsy (CP) often experience extreme poverty, compounded by limited livelihood opportunities and the added demands of caregiving, which further restrict their ability to earn an income. Targeted livelihood interventions may help improve their economic well-being. This study aimed to assess the cost-effectiveness of livelihood interventions to improve household incomes of ultra-poor families of children with CP in rural Bangladesh.

### Method

This was a mixed-methods study utilising a subgroup of a pragmatic, open-label, cluster randomised controlled trial (RCT). This subgroup was part of the "Supporting People in Extreme Poverty with Rehabilitation and Therapy (SUPPORT CP)" trial (ACTRN12619001750178), which was implemented in three rural subdistricts of Sirajganj district, Bangladesh. This RCT involved 251 children across three arms— integrated microfinance-based livelihood and community-based rehabilitation (IMCBR), community-based rehabilitation (CBR), and care-as-usual. We investigated

**Data availability statement:** The data used in this study is not publicly available due to restrictions imposed by the ethical review board, as it contains potentially identifiable or sensitive patient information. However, queries can be directed to CSF Global via email at info@csf-global.org or by phone at +88-02-55040839.

**Funding:** This trial was funded by the Research Foundation of Cerebral Palsy Alliance (PG02218—Support CP Trial) and international funding from CSF Global Bangladesh and awarded to Prof Gulam Khandaker. The funders had no role in study design, data collection and analysis, decision to publish, or preparation of the manuscript.

**Competing interests:** The authors have declared that no competing interests exist.

80 children with CP whose parents received an IMCBR program as part of the SUPPORT CP trial. Additionally, in-depth interviews were conducted with 21 participants from the IMCBR arm. Descriptive statistics to depict respondent characteristics and the average return on investment (ROI) were calculated to evaluate the most cost-effective livelihood support. Furthermore, thematic analysis was performed with the interview data to explore the advantages and disadvantages of different livelihood products.

## Results

The parents/caregivers of included children with CP were given five forms of livelihood support: Chickens (n = 3, 15 for each), Sewing machine (n = 11, 1 for each), Ghee making utensils (n = 1, 1 for each), Lamb (n = 7, 2 for each), and Goat (n = 59, 2 for each). The average cost of livelihood intervention per family was 65 · 9 USD. The net return on investment after 12 months was 59.0% for lamb, 70.0% for ghee-making tools, 24.0% for goat, 34.0% for sewing machines, and −25.0% for chicken. Lambs proved to be advantageous due to their sustainability, minimal space requirements, and disease resistance.

## Conclusions

This study suggests that the provision of lambs as livelihood support is the most effective intervention for empowering ultra-poor families with CP in Bangladesh. This experience can potentially enhance the well-being of ultra-poor families in Bangladesh and other low- and middle-income countries.

## Introduction

Disability remains an enduring concern in low- and middle-income countries (LMICs), where an estimated 80% of the global population of 1.3 billion individuals with disabilities resides [1]. Notably, these numbers are experiencing rapid growth due to an increased survival rate among individuals with disabilities [2]. These individuals predominantly rely on their families to sustain their basic livelihoods, as social support from both governmental and non-governmental organizations falls short [2]. Alarmingly, their participation in the workforce is substantially low at approximately 36%, in stark contrast to the 60% employment rate among individuals without disabilities [3]. This discrepancy significantly impacts their well-being and quality of life, a matter of paramount importance within the spheres of public health and social welfare [4]. Furthermore, the situation is compounded by the specific challenges posed by cerebral palsy (CP), a non-progressive disorder characterized by motor impairments [5]. Its effects ripple through the immediate family members due to the escalating demands for healthcare services and day-to-day care. These demands often impose an overwhelming burden, underscoring the critical need for sustained, long-term support systems [6,7].

Livelihood interventions hold the potential to elevate socioeconomic conditions and improve the overall quality of life for individuals with disabilities and their families. To date, a total of nine interventions have been examined across seven LMICs: Bangladesh, India, Nigeria, Ethiopia, Brazil, China, and Vietnam [8]. These interventions encompass a spectrum of strategies, including skills development, waged employment, financial services, social protection, and health and rehabilitation initiatives [4,8]. Their effectiveness has been demonstrated in enhancing skill acquisition for the workplace, facilitating access to the job market, and fostering employment within both formal and informal sectors [9]. However, the attainment of sustained employment remains a challenge, often due to the specific circumstances of the individuals or the nature of their disabilities [9,10]. In such instances, fostering increased access to microfinance institutes to facilitate engagement in self-employment emerges as a potential solution [8]. This approach could address the unique needs and capabilities of individuals with disabilities, allowing them to actively participate in self-sustaining economic activities. Nonetheless, critical knowledge gaps persist in our comprehension of how best to support individuals with disabilities in achieving self-employment. Currently, no conclusive evidence exists regarding the most effective form of microfinance for promoting self-employment among this marginalized population. Closing these gaps is essential to fostering more inclusive and sustainable livelihood opportunities for individuals with disabilities. Therefore, this study aimed to explore the cost-effectiveness of livelihood interventions to improve household incomes of ultra-poor families of children with CP in rural Bangladesh.

Neurodevelopmental disorders are the leading cause of childhood disability globally, with cerebral palsy (CP) being the most common among them [11,12]. This is further higher in low- and middle-income countries (LMICs) such as Bangladesh, as like other form of disability, compared to high-income countries [13,14]. In Bangladesh, the population-based prevalence of CP has been estimated to be 3.4 per 1000 children [15]. A majority of these children (68.2%) experience moderate to severe gross motor impairments, which hinder their ability to perform everyday activities [16]. As such, these individuals predominantly rely on their families to sustain their basic needs. In most cases, at least one dedicated caregiver- typically the mother, supported by other family members- is required to assist with day-to-day functioning, as social support from both governmental and non-governmental sectors remains insufficient [17,18]. One of the most pressing problems is the limited access to rehabilitation services for children with CP in Bangladesh. When such services are available, they are often prohibitively expensive, making them inaccessible to most families due to widespread economic constraints [19]. As a result, families urgently need affordable, accessible rehabilitation services and stronger institutional support to manage the long-term care and well-being of children with CP. However, this support is lacking in LMICs, including Bangladesh.

Such dependency of children with CP on their caregivers for day-to-day activities places the caregivers in disadvantaged positions, including a significant reduction in their workforce participation compared to the general population [20]. This has a considerable impact on their economic stability, directly affecting the caregivers' well-being and quality of life [4], and indirectly influencing the well-being of children with CP by limiting the caregivers' income, which could otherwise support their increasing healthcare needs [6,7]. Addressing these challenges is therefore a priority to ensure the well-being of children with CP, their caregivers, and their extended families.

Livelihood interventions have been found effective in improving the socioeconomic conditions and overall quality of life of disadvantaged populations, including people with disabilities. To date, a total of nine livelihood interventions have been examined across seven LMICs: Bangladesh, India, Nigeria, Ethiopia, Brazil, China, and Vietnam [8]. These interventions encompass a spectrum of strategies, including skills development, waged employment, financial services, social protection, and health and rehabilitation initiatives [4,8]. Their effectiveness has been demonstrated in enhancing skill acquisition for the workplace, facilitating access to the job market, and fostering employment within both formal and informal sectors [9]. However, these may not be a viable solution for caregivers of children with CP due to their limited ability to engage in formal employment [9,10]. In such cases, supporting caregivers with livelihood interventions to promote self-employment could be an effective solution, as it would enable them to care for their dependents while actively participating

in sustainable economic activities [8]. However, despite this potential, there are critical knowledge gaps regarding how best to support caregivers in creating self-employment and which types of livelihood interventions are most effective. Closing these gaps is essential to fostering more inclusive and sustainable livelihood opportunities for caregivers of individuals with disabilities. Therefore, this study aimed to explore the cost-effectiveness of livelihood interventions to improve household incomes of ultra-poor families of children with CP in rural Bangladesh.

## Methods

### Design and setting

This study employed a mixed-methods design, integrating quantitative and qualitative approaches to provide a comprehensive understanding of the effectiveness of livelihood interventions. The quantitative component consisted of subgroup analyses of a cluster randomised controlled trial (RCT) entitled Supporting People in Extreme Poverty with Rehabilitation and Therapy (SUPPORT CP) [7,21]. A sub-sample of participants from the quantitative study was selected for qualitative interviews to gain deeper insights. The SUPPORT CP trial was a programmatic and open-level cluster RCT involving 251 children with CP and their primary caregivers across three arms— integrated microfinance-based livelihood and community-based rehabilitation (IMCBR), community-based rehabilitation (CBR), and care-as-usual. This trial aimed to explore the efficacy of an integrated micro-finance/livelihood and community-based rehabilitation (IMCBR) programme in improving health-related quality of life and motor function of children with CP living in ultra-poor families. The trial was implemented in three rural sub-districts: Shahjadpur, Belkuchi, and Ullahpara, all located within the Sirajganj district of Bangladesh. Baseline and end-line assessments were conducted in December 2019 and February 2021, respectively, and were considered the quantitative approach. All the qualitative data were collected in July 2022. The following inclusion criteria were considered for participant selection: (1) being the primary caregiver of a child with CP aged five years or younger, and (2) belonging to a family categorised as ultra-poor based on the World Bank criterion of less than US$1.90 per day per capita income [22]. It is important to note that the World Bank updated the cut-off value for classifying ultra-poor families to US$2.15 per day in September 2022 [23]. However, since our data were collected prior to this change, we used the earlier cut-off values. However, since our data collection was conducted prior to change, we have used the earlier cut-off value in our analysis. Before quantitative and qualitative data collection, informed written consent was obtained from the participants. Only the corresponding author had access to information to identify the participants during and after the study trial.

### Participants and recruitment of the study

For this study, we analysed data of primary caregivers of the IMCBR arm, who received a microfinance-based livelihood program, and their children with CP, who received a community-based rehabilitation program. We also conducted qualitative interviews of a subgroup of 21 participants using a semi-structured questionnaire. The basis for such selection was proactive participation in the program activities and providing data, ensuring that the data represented each livelihood asset receiver category. The interviews were conducted one-on-one by a research physician and held in the participant's house.

### Livelihood support

A structured one-on-one assessment of the primary caregivers was conducted by a research physician and field coordinator. Based on this assessment, further consultation with the caregivers, regarding their preferences, existing livelihood activities, local environment, and capacity, livelihood assets were chosen for the caregivers as a microfinance-based livelihood program. The selected assets included goats, lambs, sewing machines, ghee-making utensils, and chicken.

After asset distribution, a veterinary surgeon from the local government livestock department provided a three-day hands-on training for participants on scientific livestock rearing practices. This training covered construction of animal shelters, preparation of balanced diets, identification and management of common diseases, and basic veterinary care. In parallel, other local experts delivered relevant training tailored to non-livestock livelihood activities, enhancing participants' skills and knowledge in income-generating activities. Pre- and post-training evaluations were conducted to identify specific knowledge gaps. Follow-up training sessions were organized to address these gaps, ensuring participants could manage their assets effectively to generate income.

During the trial, caregivers were expected to participate in weekly meetings to discuss progress, raise concerns, and explore locally appropriate solutions. However, these meetings were discontinued after 2.5 months due to the COVID-19 pandemic. To ensure continuity and effective monitoring, trained field coordinators conducted bi-monthly home visits to assess the status of livelihood activities, review financial records, and provide ongoing technical support. During these visits, participants reported income and expenditure related to the livelihood assets they received. To enhance accuracy and credibility, these self-reported financial statements were cross verified with prevailing local market prices and, where possible, triangulated through observation of production outputs (e.g., livestock, eggs, ghee). Outcomes were independently assessed by a blinded assessor at baseline, 6 months, and 12 months. The total return obtained by each participant was calculated as the cumulative net income generated over the 12-month period, based on the verified financial reports, and was considered the primary outcome measure.

## Statistical and cost-effectiveness analysis

We employed descriptive statistics to delineate the characteristics of the respondents. We calculated the average ROI by comparing the market price of the provided livelihood support at baseline (January 2020) with the market price of the support and its output at the end line (December 2020).

To calculate the ROI, we employed the following formula: $\dfrac{\text{Gain from Investment} - \text{Cost of Investment}}{\text{Cost of Investment}} * 100.$

The net gain from the investment was determined by subtracting the initial cost of the investment from the total benefits obtained over the investment period. These benefits included direct financial gains attributable to the intervention or investment under study. It is important to note that the Government of Bangladesh provides veterinary services and compulsory animal vaccinations free of charge. Therefore, these costs were not included in the ROI analysis. We conducted cost-benefit analyses as sensitivity analysis to explore the benefits from different types of investment considering the conditions How changes in certain factors (like unit price or quantity) impact the overall profitability was also explored. All statistical analyses were carried out using Stata version 15.2.

## Qualitative data analysis

The qualitative data obtained from interviews were analyzed using thematic analysis. Thematic analysis followed Braun and Clarke's (2006) six-phase framework, which included familiarizing the data, generating initial codes, searching for themes, reviewing themes, defining and naming themes, and producing the final report.

## Ethical considerations

The trial received ethical approval from the Human Research Ethics Committee of the Asian Institute of Disability and Development (approval number: Southasia-hrec-2019-5-03), and Bangladesh Medical Research Council (approval number: BMRC/NREC/2016–2019/251). Additionally, the trial was registered with the Australian New Zealand Clinical Trials Registry (registration number: ACTRN12619001750178). Primary caregivers of children with CP provided written informed consent for their participation.

## Results

### Composition and the Background characteristics of the participants

Among the participants, the majority of the mothers of children with CP (48.4% [n = 39]) had secondary education and were housewives (91.3% [n = 73]) (Table 1). In contrast, 35% (n = 28) of fathers had secondary education, and 65.8% (n = 52) were employed in blue-collar occupations. In terms of the number of household members, 71.2% (n = 65) of the participants' families had between 5 and 13 members, while 76.2% (n = 61) of families had only one earning member, and 23.8% (n = 15) had two or more earning members. Furthermore, approximately 29.8% (n = 23) of the families had a monthly income of less than 7,000 BDT (equivalent to USD 82.57).

**Table 1. Background characteristics of the respondents.**

| Characteristics | n (%) |
|---|---|
| Mothers' education | |
| No education | 6 (7 · 5) |
| Up to primary education | 31 (38 · 8) |
| Up to secondary education | 39 (48 · 4) |
| Higher education | 4 (5 · 0) |
| Mothers' occupation | |
| Housewives | 73 (91 · 3) |
| Others | 7 (8 · 7) |
| Father's education | |
| No education | 21 (26 · 3) |
| Up to primary education | 28 (35 · 0) |
| Up to secondary education | 28 (35 · 0) |
| Higher education | 3 (3 · 8) |
| Father's occupation | |
| Agriculture | 12 (15 · 2) |
| Blue color worker | 52 (65 · 8) |
| White color worker | 2 (2 · 5) |
| Pink color worker | 10 (12 · 7) |
| Others | 3 (3 · 8) |
| Number of household members | |
| ≤4 | 45 (56 · 2) |
| 5-6 | 20 (25 · 0) |
| 8-13 | 15 (18 · 8) |
| Number of earning member in the family | |
| Only 1 people | 61 (76 · 2) |
| 2 or more | 19 (23 · 8) |
| Monthly family income | |
| ≤BDT 7000 (USD 82.6) | 23 (29 · 8) |
| BDT 7001–9000 (USD 82.6–106.1) | 8 (10 · 4) |
| BDT 9001–12000 (USD 106.2–141.5) | 16 (20 · 8) |
| BDT 12001–15000 (USD 141.6–177.0) | 18 (23 · 4) |
| BDT 15001–100000 (USD 177.0–1179.6) | 12 (15 · 6) |

Note: 1 USD = 84 · 78 BDT (January 2020)

### Livelihood support and their cost-effectiveness through return on investment

The average cost of livelihood intervention per family was 65 · 9 USD. Primary caregivers of children with CP received five forms of livelihood support: Chickens (n = 3, 15 for each), Sewing machine (n = 11, 1 for each), Ghee making utensils (n = 1, 1 for each), Lamb (n = 7, 2 for each), and Goat (n = 59). The ROI for lamb was 59%, for goat was 24%, for ghee-making utensils was 70%, for a sewing machine was 34% and for chicken was −25% (Table 2). Among the three households provided with chickens, two indicated that the chickens had been eaten, lost, or perished due to inadequate care. Conversely, comparatively higher returns on investment were reported among households that received sewing machines, ghee-making utensils, and goats. Notably, the return figures for households with lambs were approximately two times greater than the return from households provided with goats or sewing machines.

### Lamb

The base sell price of lamb was BDT 6643 (USD 78.3) in December 2020, and we considered the range of prices between BDT 3,000 and BDT 11,000 (USD 35.4 and 129.7) per lamb to examine the sensitivity of price. We had 14 lambs in our intervention study, so we considered a range of lambs between 7 and 15 to examine the sensitivity of quantity (S1 Table of the S1 File). For BDT 3,000 (USD 35.4) price per unit: At 7 units, the profit was BDT 875 (USD 10.3); at 15 units, it increased to BDT 1,875 (USD 22.1). For BDT 5,000 (USD 58.9) price per unit: At 7 units, the profit was BDT 14,875 (USD 175.4), and at 15 units, it increased to BDT 31,875 (USD 375.8). For BDT 7,000 (USD 82.5) price per unit: At 7 units, the profit was BDT 28,875 (USD 340.5), and at 15 units, the profit was BDT 61,875 (USD 729.6). For BDT 9,000 (USD 106.1) price per unit: At 7 units, the profit was BDT 42,875 (USD 505.5), and at 15 units, the profit was BDT 91,875 (USD 1083.3). For BDT 11,000 (USD 129.7) price per unit: At 7 units, the profit was BDT 56,875 (USD 670.6), and at 15 units, the profit was BDT 121,875 (USD 1437.0) Higher unit prices and greater numbers of units led to significantly higher profits. The sensitivity analysis showed a positive ROI between lower and higher ranges of prices and quantities. So, the intervention model could have been sustainable even if the market prices had fluctuated between BDT 3000 and BDT 11,000 (USD 35.3 and 129.7) or higher. In addition, as few as 7 lambs could have bought positive ROI between the lowest and highest price range. However, if the price had gone below BDT 3,000, (USD 35.4) the model would not have been sustainable, but chances were very low.

**Table 2. Types of livelihood support given to the respondents and return on investment. Notes: 1 USD = 84 · 78 BDT (January 2020); 1 USD = 84 · 81 BDT (December 2020).**

| Types of Livelihood Support Provided (a) | N (number of beneficiaries) | Per Unit Price in BDT | Purchase cost January 2020 (In BDT & USD) | Cost of total loss or damage (In BDT & USD) | Yearly total Maintenance cost (In BDT & USD) | Total costs (In BDT & USD) | Total cost per unit (IN BDT & USD) | Total Price in December 2020 (in BDT & USD) | Total income per unit (IN BDT & USD) | Net Revenue December 2020 (in BDT & USD) | The net return on investment (%) |
|---|---|---|---|---|---|---|---|---|---|---|---|
| Chickens | 45 (n = 3, 15 for each) | 245 | 11025 (130 · 0) | 6125 (72.3) | 0 | 17150 (202.29) | 381 (4 · 5) | 14400 (169.8) | 320 (3.8) | −2750 (−32.4) | −25% |
| Sewing Machine | 11 (n = 11, 1 for each) | 5600 | 61600 (726 · 6) | 4000 (47.2) | 9225 (108.8) | 74825 (882.6) | 6802 (80.2) | 95920 (1131.0) | 8720 (102.8) | 21095 (248.7) | 34% |
| Ghee making tool | 1 (n = 1, 1 set of utensils) | 5000 | 5000 (59 · 0) | 500 (5.9) | 500 (5.9) | 6000 (70.8) | 6000 (70 · 8) | 9500 (112.0) | 9500 (112.0) | 3500 (41.3) | 70% |
| Lamb | 14 (n = 7, 2 for each) | 2875 | 40250 (474 · 8) | 5750 (67.8) | 23352 (275.4) | 69352 (818 · 0) | 4954 (58.4) | 93000 (1096.6) | 6643 (78.3) | 23648 (278.8) | 59% |
| Goat | 118 (n = 59, 2 for each) | 2835 | 334530 (3945 · 9) | 119070 (1404 · 5) | 98280 (1159 · 2) | 551880 (6509 · 6) | 4677 (55 · 2) | 633000 (7463.7) | 5364 (63.3) | 81120 (956.5) | 24% |

### Sewing machine

The sensitivity analysis for sewing machines examined different income ranges per sewing machine (BDT 5,000 to BDT 25,000, USD 59.0 to 294.8). In the trial, as we provided 11 sewing machines, we considered the number of sewing machines from 5 to 15 for sensitivity analyses (S2 Table of the S1 File). At BDT 5,000 (USD 59.0) price per unit: All entries were in red, indicating a loss ranging from BDT 3,000 to BDT 9,000 (USD 35.4 to 106.1). At BDT 10,000 (USD 118.0) price per unit: Profit ranged from BDT 22,000 to BDT 66,000 (USD 259.4 to 778.2). At BDT 15,000 (USD 176.9) price per unit: Profit ranged from BDT 47,000 to BDT 141,000 (USD 554.1 to 1662.5). At BDT 20,000 (USD 235.8) cost per unit: Profit ranges from BDT 72,000 to BDT 216,000 (USD 849.0 to 2546 · 9). At BDT 25,000 (USD 294.8) price per unit: Profit ranged from BDT 97,000 to BDT 291,000 (USD 1143.7 to 3431.2). All scenarios resulted in a loss at the lowest unit price (BDT 5,000 or USD 59.0). However, the investment became highly profitable as the income per unit increased, not less than BDT 5000 (USD 59.0).

### Goat

The sensitivity analysis for goats examined prices per goat (BDT 5,000 to BDT 13,000, USD 59.0 to 153.9) and the number of goats (70–120). In December 2020, the goat's base sell price was BDT 5364 (USD 63.3), and we considered a range of prices between BDT 5,000 and BDT 13,000 (USD 59.0 to 153.9) per goat to examine the price sensitivity (S3 Table of the S1 File). We had 118 goats in our intervention study, so we considered a range of goats between 70 and 120 to examine the sensitivity of quantity. At BDT 5,000 (USD 59.0) price per goat: Profit ranged from BDT 151,550 to BDT 259,800 (USD 1786.9 to 3063.3). At BDT 7,000 (USD 82.5) price per goat: Profit ranged from BDT 291,550 to BDT 499,800 (USD 3437.7 to 5893.2). At BDT 9,000 (USD 106.1) price per unit: Profit ranged from BDT 428,750 to BDT 735,000 (USD 5055.4 to 8666.4). At BDT 11,000 (USD 129.7) price per unit: Profit ranged from BDT 568,750 to BDT 975,000 (USD 6706.2 to 11496.3). At BDT 13,000 (USD 153.9) price per unit: Profit ranged from BDT 708,750 to BDT 1,215,000 (USD 8356.9 to 14326.1). Higher unit prices and greater numbers of units led to significantly higher profits. The sensitivity analyses indicated that the intervention model remained sustainable with a positive ROI across various prices and quantities. Specifically, the model was viable even if market prices had fluctuated between BDT 5,000 and BDT 13,000 (USD 59.0 to 153.9) or higher. Additionally, as few as 70 goats were needed to maintain a positive ROI within this price range. However, if prices had dropped below BDT 5,000, (USD 59.0) the model's sustainability would have been compromised, although this scenario was considered unlikely.

### Ghee making utensils

The sensitivity analyses for the ghee-making utensils examined different income generated per unit (BDT 3,000 to BDT 10,000 USD 35.4 to 117.9) and the number of units of utensils (S4 Table of the S1 File). We considered units 1 to unit 6 as we had provided ghee-making utensils to a single family (n = 1) in the trial. At BDT 3,000 (USD 35.4) price per unit: Loss ranged from BDT 2,000 to BDT 12000 (USD 23.6 to 141.5) for 6 units. At BDT 5,000 (USD 59.0) price per unit: No loss or profit at any number of units. At BDT 7,000 (USD 82.5) price per unit: Profit ranged from BDT 2,000 to BDT 12,000 (USD 23.6 to 141.5). At BDT 9,000 (USD 106.1) price per unit: Profit ranged from BDT 4,000 to BDT 24,000 (USD 47.1 to 283.0). At BDT 10,000 (USD 117.9) price per unit: Profit ranged from BDT 5,000 to BDT 30,000 (USD 59.0 to 353.7). At BDT 5,000 (USD 59.0) per unit, there was no loss or profit. The higher income per unit resulted in positive profits, indicating a direct relationship between the unit and profit. To ensure profit, making the product and marketing was crucial. The sensitivity analyses for lamb, sewing machines, goats, and ghee-making tools revealed that higher unit prices generally led to higher profits. Investments in sewing machines and ghee-making tools could have resulted in losses if the per-unit income was too low. For lamb and goats, profits increased significantly with higher unit prices and greater quantities. Despite the small scale, the ghee-making tool showed the highest net return of 70%, suggesting it was a highly profitable investment Table 3.

**Table 3. Lessons learned about different livelihood activities and the key to success.**

| Types of livelihood support provided | Advantage | Disadvantage | Things to avoid | Key to success |
|---|---|---|---|---|
| Chickens | In a free-range system, they collect food from the surroundings, reducing food expenditure.<br>The price of the local breed and free-range chickens are comparatively lot higher than the hybrid chickens ·<br>In a year, a local breed hen provides approximately 30–45 chickens and if these chickens are maintained properly, a decent amount of money can be flowed on a regular basis after meeting the family's protein demand with egg and chicken. | Need to provide a safe and dedicated protected shelter house to keep the chickens safe from other animals like foxes or dogs.<br>Could be a health hazard if not cleaned or maintained properly.<br>Disease prone and low immunity against diseases if not properly vaccinated. | Rearing in open, unprotected, and unhygienic shelter.<br>Late or incomplete vaccination.<br>Not providing any extra food supplements during the reproductive period. | Providing proper and safe shelter ·<br>Timely vaccination and regular check-up by the veterinary doctor for any sickness.<br>Added extra granular foods regularly and multivitamins if needed.<br>Proper marketing of eggs and chickens in the local village market. |
| Sewing Machine | It's a fixed asset.<br>Usually, a long-life cycle if maintained properly.<br>Flexible working hours.<br>Repairing is easy and usually cheap. | Enables sound pollution which can cause inconvenience to family and neighbors.<br>Income depends upon the availability of work ·<br>Needs separate space inside the home to work properly. | 1. If there is not sufficient work from neighboring areas or any small/large scale industry.<br>2 · If other members of the family are not helpful in carrying in and out the sewed product, it is very tough to take orders and timely delivery of the product. | Maintaining regular liaison with the local loom industries and local customers.<br>Maintaining regularity and availability and the quality of the work. |
| Ghee making tool | Can use utensils according to the need.<br>Low maintenance.<br>Reusable, durable, and can use continuously for a minimum of 5–8 years. | Market demand is low due to high prices and the availability of alternative low-cost products.<br>Need regular customers or shops to sell the product.<br>The raw materials to produce ghee are expensive, and the price fluctuates according to the market, so a fixed price for the product can't be set. | As a small and competitive market, the inability to identify the right distribution channel would make the business unprofitable.<br>The presence of unhygienic and artificial products makes the market for organic and original ghee unstable. | Must maintain a regular customer base.<br>Online marketing is a good solution for getting regular customers.<br>Need to maintain competitive prices and ensure the original organic quality. |
| Lamb | In a small area, these animals can be reared.<br>More resistant to diseases.<br>Mainly fed on locally grown crops and plants, therefore easy to rear · Need to provide minimal granular foods if green grass and plants are available.<br>Highly productive. A female lamb gives birth twice a year and has at least 1–3 babies at each birth.<br>Farmers can get wool, milk, and meat if reared properly.<br>They have a different market value all year long because their fat content is lower, and their meat chunks are higher. | The market price of a full-grown lamb is not as high as a goat's.<br>Require at least a flock of lamb to make a profit like other livestock, e.g., goats and cows.<br>The market for selling lamb milk is not well-established. | Not providing a balanced diet, salt, and extra food during pregnancy.<br>Unhygienic shelter and lack of care.<br>Improper vaccination. | Taking care regularly and maintaining a healthy shelter.<br>Providing green grass regularly along with granular foods.<br>Regular medication and checkups during pregnancy.<br>Selling the animals in a local hut gives a better monetary value. |

*(Continued)*

**Table 3.** (Continued)

| Types of livelihood support provided | Advantage | Disadvantage | Things to avoid | Key to success |
|---|---|---|---|---|
| Goat | Need small facilities to rear.<br>Mainly fed on locally grown crops and plants, therefore easy to rear · Need to provide minimal granular foods if green grass and plants are available.<br>Highly productive. A female goat gives birth twice a year and has at least 1–4 babies at each birth.<br>Goat milk has a different market and extra value, so it can be an additional income source for farmers. | Disease-prone, especially in the winter and rainy seasons, are more problematic for goats.<br>Need to give vaccines on time and medication for any disease.<br>Need a spacious place to stay at night, and shelter is needed to be open with full of light and air.<br>Need a person to stay along with while rearing because goats tend to eat everything that is available in their way.<br>Goes to nearby houses if not checked properly and eats plants and other things.<br>Easy target to the rabid and street dogs in regular areas and in the rearing fields. | Not providing a balanced diet, salt, and extra food.<br>Unhygienic shelter and lack of care.<br>Improper vaccination and keeping the babies in outside during rain and chilling weather.<br>Selling from home rather than in the village market. | Hygienic shelter · Providing extra granular foods during reproductive age and pregnancy.<br>Providing green grass regularly.<br>Giving the vaccines on time.<br>Selling goats in the market, not from home.<br>Regular check-ups and treatment for disease.<br>Regular artificial insemination with better breed. |

## Discussions

To the best of our knowledge, this is the first study conducted in a LMIC to compare the effectiveness of various livelihood interventions aimed at improving the wellbeing of caregivers of children with CP. Among the interventions provided—such as chickens, sewing machines, ghee-making utensils, lambs, and goats—the findings indicate that allocating resources toward providing lambs yields the highest return. Therefore, offering lambs represents a strategically impactful approach for supporting ultra-poor households in Bangladesh, particularly families that include persons with disabilities. This insight carries important implications for both governmental and non-governmental organizations in designing field-level programs to support ultra-poor families affected by disability.

Families of children with CP in rural Bangladesh face unique challenges due to the combined burden of poverty and disability. In this study, 91.3% of the primary caregivers were mothers, most of whom were homemakers with limited education—only 38.8% completed primary education, and 7.5% had no formal education. The vast majority of families (76.2%) had only one income earner, typically engaged in low-paid agricultural or blue-collar work (81%). Over half of the households had four or more members, intensifying financial strain. These circumstances severely limited caregivers' ability to prioritize rehabilitation for their children. Livelihood interventions tailored to such contexts—such as small livestock or home-based enterprises—offered practical, income-generating opportunities that aligned with caregivers' existing responsibilities [24]. This study highlights the importance of disability-inclusive livelihood programs as a critical strategy to improve the well-being of families with children with CP in low-resource settings.

While evidence supports the benefits of livelihood support programs for people with disabilities in LMICs, limited guidance exists on which specific interventions are most effective. A recent systematic review encompassing 10 studies from LMICs, concluded that livelihood support programs are effective in enhancing the well-being of people with disabilities [8]. However, both the review itself and the studies it incorporated lacked clarity in identifying which types of interventions yield the greatest benefits and in explaining the mechanisms through which these benefits are achieved [25–34]. The present

study builds on this body of work by offering new insights into the relative effectiveness of specific livelihood interventions and elucidating how they contribute to improving the economic stability and well-being of families of children with CP.

This study underscores the provision of lambs as the foremost effective livelihood support for ultra-poor families in Bangladesh. Following closely behind in effectiveness were goat rearing and the utilization of ghee-making tools as the second and third most effective interventions, respectively. These findings signify a shift away from the conventional approach of one-time financial assistance, commonly employed by government and non-government organizations in Bangladesh and other LMICs [8,32]. Such financial assistance often gets rapidly depleted without fostering sustainable change [35]. In contrast, the study suggests a more sustainable approach that directly involves providing these specific forms of livelihood support.

The advantages of providing lambs as livelihood support and achieving higher returns are clear. Firstly, lambs require minimal space and can be raised by providing locally grown crops as their primary feed source. Additionally, lambs exhibit high productivity, with female lambs capable of giving birth twice a year and typically having 1–3 offspring per birth. Moreover, they tend to be more resistant to diseases compared to other livestock [36]. Consequently, rearing lambs yields higher returns, surpassing even the more commonly reared goats in Bangladesh. As a result, the market price of goats is consistently higher than that of lambs. To further enhance the profitability of lamb rearing, a focus on increasing the number of lambs raised and implementing effective marketing strategies is advisable [37]. Additionally, training caregivers on topics such as balanced diets, supplementary feeding during pregnancy, and proper vaccination practices can optimize lamb production and contribute to even greater profitability in the future. These measures can empower ultra-poor families to leverage the full potential of lamb rearing as a sustainable livelihood option, which also has the potential to transform short-term assistance into long-term empowerment, offering a promising pathway to economic resilience and improved well-being for these vulnerable communities.

The findings of this study also endorse the idea that rearing lambs and goats can be even more effective when it comes to supporting the acquisition of income-generating equipment, such as sewing machines or ghee-making utensils, which are commonly utilized in Bangladesh. The underlying reasons are that they require physical labor and the ability to maintain close contact with local markets. Furthermore, generating income from these productive tools often involves competing in local markets largely dominated by large-scale companies [38]. This competitive landscape can pose significant challenges for individuals seeking to establish themselves in the market. Therefore, while equipment-based income generation is a viable option, it demands physical effort and the ability to navigate and succeed in a competitive business environment, which may not align with the circumstances of caregivers or individuals with disabilities.

This study boasts several strengths alongside a few limitations. Its most noteworthy strength lies in the comparative analysis of returns from various livelihood programs, using data obtained from a well-structured cluster RCT. The clusters were thoughtfully selected, and samples were randomly allocated into different arms, ensuring the robustness of the study's findings. Consequently, the insights gleaned from this study can be instrumental in shaping the design of future livelihood support programs in Bangladesh and other LMICs aimed at assisting ultra-poor families with children with disabilities. However, a significant limitation arose due to the unforeseen COVID-19 pandemic, which imposed various restrictions. These restrictions impacted the planned activities intended to make livelihood support programs more effective, leading to suboptimal execution. Completing these activities adequately could have potentially further enhanced the effectiveness of livelihood support programs. Therefore, it's important to note that the reported average returns in this study may underestimate the true potential. Moreover, in calculating the average return, we considered the average market prices of the livelihood support at both the baseline and end-line. It is essential to acknowledge that the prices of these support items can vary over time and across different locations, and the Inflation rate over the period of RCT could also impact this study's findings. As a result, the average return reported in this study may not necessarily be representative in all situations. The RCT was implemented in rural areas rich in natural food resources, such as grass and leaves, which may limit the generalizability of findings to more resource-constrained urban areas. In addition, participants' background

characteristics—such as education, occupation, and other factors that could influence the effective use of received investments and the potential for better returns—were important considerations. However, we were unable to explore these factors quantitatively due to the lack of disaggregated data across the different types of livelihood support received. Nonetheless, these aspects were examined as thoroughly as possible through qualitative exploration. Despite these limitations, the findings of this study hold significant promise for developing future livelihood support programs tailored to ultra-poor families with disabilities in rural areas. The insights gained here can serve as a valuable foundation for addressing the unique challenges these vulnerable populations face in LMICs.

## Conclusions

This study delves into the effectiveness of various livelihood support options tailored for ultra-poor families with children with disabilities. Our research has revealed that, among these options, lambs emerge as the most potent livelihood support for such vulnerable families, with goats and ghee-making tools closely following in terms of effectiveness. These findings carry substantial implications, particularly regarding the future of livelihood support programs in Bangladesh and other LMICs. The prominence of lambs as a livelihood support option underscores their potential to significantly impact the well-being and economic prospects of ultra-poor families with disabilities. This revelation encourages the consideration of lambs as a central component of future support programs aimed at uplifting these marginalized communities.

## Supporting information

**S1 File.** S1 Table. Sensitivity analysis of lamb. S2 Table. Sensitivity analysis of sewing machine. S3 Table. Sensitivity analysis of goat. S4 Table. Sensitivity analysis of ghee making utensils.
(DOCX)

## Acknowledgments

We extend sincere gratitude to primary caregivers and children with CP for their valuable time and voluntary involvement in the SUPPORT CP trial. Additionally, we express heartfelt appreciation to the dedicated CSF global team in Bangladesh for their diligent efforts and support.

## Author contributions

**Conceptualization:** Mohammad Muhit, Nadia Badwai, Gulam Khandaker.

**Data curation:** Israt Jahan, Nadia Badwai, Gulam Khandaker.

**Formal analysis:** Manik Chandra Das, Mahmudul Hassan Al Imam, Israt Jahan, Delwar Akbar.

**Funding acquisition:** Gulam Khandaker.

**Investigation:** Manik Chandra Das, Mahmudul Hassan Al Imam, Israt Jahan, Gulam Khandaker.

**Methodology:** Nuruzzaman Khan, Manik Chandra Das, Mahmudul Hassan Al Imam, Delwar Akbar, Gulam Khandaker.

**Project administration:** Manik Chandra Das, Mahmudul Hassan Al Imam, Israt Jahan, Mohammad Muhit, Nadia Badwai, Gulam Khandaker.

**Resources:** Mohammad Muhit, Nadia Badwai, Gulam Khandaker.

**Software:** Nuruzzaman Khan, Manik Chandra Das, Israt Jahan.

**Supervision:** Mahmudul Hassan Al Imam, Israt Jahan, Mohammad Muhit, Nadia Badwai, Gulam Khandaker.

**Validation:** Israt Jahan, Nadia Badwai.

**Visualization:** Nuruzzaman Khan, Mohammad Muhit, Nadia Badwai.

**Writing – original draft:** Nuruzzaman Khan, Manik Chandra Das.

**Writing – review & editing:** Nuruzzaman Khan, Manik Chandra Das, Mahmudul Hassan Al Imam, Israt Jahan, Delwar Akbar, Nadia Badwai, Gulam Khandaker.

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
