## [Editor Report · Decision Letter 0]

13 Aug 2024

Dear Dr. Khandaker,

Thank you for submitting your manuscript to PLOS ONE. After careful consideration, we feel that it has merit but does not fully meet PLOS ONE’s publication criteria as it currently stands. Therefore, we invite you to submit a revised version of the manuscript that addresses the points raised during the review process.

We look forward to receiving your revised manuscript.

Kind regards,

Mohammad Shafiqul Islam

Academic Editor

PLOS ONE

Journal Requirements:

2. Thank you for stating the following financial disclosure: This trial was funded by the Research Foundation of Cerebral Palsy Alliance (PG02218—Support CP Trial) and international funding from CSF Global Bangladesh and awarded to Prof Gulam Khandaker.   

3. Thank you for stating the following in the Acknowledgments Section of your manuscript: We extend sincere gratitude to primary caregivers and children with CP for their valuable time and voluntary involvement in the SUPPORT CP trial. Additionally, we express heartfelt appreciation to the dedicated CSF global team in Bangladesh for their diligent efforts and support. This trial was funded by the Research Foundation of Cerebral Palsy Alliance (PG02218—Support CP Trial) and international funding from CSF Global Bangladesh. The authors declare that there is no conflict of interest. Research data can be accessed by contacting the corresponding author.

Please remove any funding-related text from the manuscript and let us know how you would like to update your Funding Statement. Currently, your Funding Statement reads as follows: This trial was funded by the Research Foundation of Cerebral Palsy Alliance (PG02218—Support CP Trial) and international funding from CSF Global Bangladesh and awarded to Prof Gulam Khandaker.  

4. We notice that your supplementary tables are included in the manuscript file. Please remove them and upload them with the file type 'Supporting Information'. Please ensure that each Supporting Information file has a legend listed in the manuscript after the references list.

5. We note you have included a table to which you do not refer in the text of your manuscript. Please ensure that you refer to Table 1 in your text; if accepted, production will need this reference to link the reader to the Table.

Additional Editor Comments:

Please see attached file

---

## [Author Response · Author response to Decision Letter 1]

26 Aug 2024

We have uploaded an word file, where we have addressed all the comments.

---

## [Decision Letter · Decision Letter 1]

23 Mar 2025

Dear Dr. Khandaker,

Thank you for submitting your manuscript to PLOS ONE. After careful consideration, we feel that it has merit but does not fully meet PLOS ONE’s publication criteria as it currently stands. Therefore, we invite you to submit a revised version of the manuscript that addresses the points raised during the review process.

We look forward to receiving your revised manuscript.

Kind regards,

Najmul Hasan, PhD

Academic Editor

PLOS ONE

Reviewers' comments:

Reviewer's Responses to Questions

**Comments to the Author**

Reviewer #1: All comments have been addressed

Reviewer #2: (No Response)

2. Is the manuscript technically sound, and do the data support the conclusions?

Reviewer #1: Yes

Reviewer #2: Partly

3. Has the statistical analysis been performed appropriately and rigorously?

Reviewer #1: Yes

Reviewer #2: N/A

4. Have the authors made all data underlying the findings in their manuscript fully available?

Reviewer #1: Yes

Reviewer #2: Yes

5. Is the manuscript presented in an intelligible fashion and written in standard English?

Reviewer #1: Yes

Reviewer #2: Yes

Reviewer #1: This is a very interesting study about children with cerebral palsy. However, there are the following issues that need further clarification:

1. The severity of CP patients, whether they can live independently, and the basic expenses for treatment and rehabilitation.

2.How many family members were needed to take care of in their daily lives?

3. What are the urgent problems that these patients and families need to solve?

Reviewer #2: Thank you for the editor's invitation. The authors present in this manuscript the differential outcomes of various support modalities for families of children with cerebral palsy. This study demonstrates innovative elements and could provide valuable insights for developing countries and impoverished regions. However, prior to publication, I have the following suggestions to offer.

1. In lines 76-80, it is recommended that the authors first provide a comprehensive introduction to the clinical characteristics of cerebral palsy, followed by an in-depth discussion on how this condition exacerbates the socioeconomic burden on families from low-income backgrounds.

2. More detailed inclusion and exclusion criteria for the study samples should be added.

3. In lines 143-144, "After distributing the livelihood assets, relevant training sessions were organized to develop participants' skills and knowledge." Please provide more detailed information about these training sessions, such as whether the proficiency levels of each participant are comparable.

4. Please supplement the methodology section with the methodology for sensitivity analysis.

5. How is the total return obtained by participants during the investment period determined, and how reliable is it? The authors need to offer a more comprehensive methodological explanation to enhance readers' confidence in the accuracy of the results.

6. Does the educational level and occupation of parents in different families indirectly affect the rate of return on investment?

7. The credibility of results obtained solely based on differences in the rate of return on investment is relatively low, it is necessary to employ more statistical methods, such as linear regression, to determine whether there is significance.

8. Conducting subgroup analyses based on participants' backgrounds is indeed necessary.

9. The line numbers disappeared after the tenth page.

10. In the section "Lessons learned about different livelihood activities and the key to success," the conclusions are partially subjective and should be supported by an appropriate amount of references.

11. In the discussion section, "To the best of our knowledge, this study compares the efficacy of various livelihood support interventions to determine the most effective approach, is a pioneering effort within LMIC settings." Please explicitly state that this is the most effective approach for improving the well-being of families with children affected by cerebral palsy.

12. The study primarily focuses on the livelihoods of families with children affected by cerebral palsy. However, in the discussion section, there is insufficient analysis and discussion regarding this specific population group.

**Do you want your identity to be public for this peer review?** For information about this choice, including consent withdrawal, please see our Privacy Policy

Reviewer #1: **Yes: ** Xiuyu Du

Reviewer #2: **Yes: ** Xuanjie Chen

---

## [Author Response · Author response to Decision Letter 2]

21 May 2025

We have uploaded a MS word file providing point by point response to each of the reviewer's comment.

---

## [Editor Report · Decision Letter 2]

4 Jun 2025

Cost-effectiveness of livelihood interventions for families of children with cerebral palsy in rural Bangladesh

PONE-D-24-27317R2

Dear Dr. Khandaker,

We’re pleased to inform you that your manuscript has been judged scientifically suitable for publication and will be formally accepted for publication once it meets all outstanding technical requirements.

Kind regards,

Najmul Hasan, PhD

Academic Editor

PLOS ONE

Additional Editor Comments (optional):

Following a comprehensive review, your paper has been acknowledged for its substantial contribution to the field of health sciences. The study offers valuable insights into the cost-effectiveness of livelihood interventions, and I am confident with this current version that it will have a significant impact on policy discussions, as well as practical applications aimed at improving the lives of children with cerebral palsy and their families in rural Bangladesh